# Effect of Polyethylene Glycol on Preparation of Magnesium Hydroxide by Electrodeposition

**DOI:** 10.3390/ma15093278

**Published:** 2022-05-03

**Authors:** Zhichun Cui, Yuezhong Di, Jianping Peng, Yaowu Wang, Naixiang Feng

**Affiliations:** School of Metallurgy, Northeastern University, Shenyang 110819, China; cuizc0121@163.com (Z.C.); pengjp@smm.neu.edu.cn (J.P.); wangyw@smm.neu.edu.cn (Y.W.); fengnaixiang@163.com (N.F.)

**Keywords:** magnesium hydroxide, polyethylene glycol, electrodeposition, surfactant

## Abstract

The current research focuses on the mechanism of the surfactant polyethylene glycol (PEG) in the preparation of magnesium hydroxide by electrolysis of a salt lake bischite aqueous solution. The samples were analyzed by a scanning electron microscope (SEM), X-ray diffraction (XRD), Fourier transform infrared spectrometer (FT-IR) and laser particle size analyzer. The characterization results show that PEG plays an important role in adjusting the growth mode and morphology of Mg(OH)_2_ crystals. The ether group of the PEG molecular chain and the hydroxyl group of Mg(OH)_2_ can be combined by a hydrogen bond, which provides a "template" for the growth of Mg(OH)_2_. At the same time, the difference in growth mode and morphology will also affect the economic performance of electrolytic reactions. When the PEG content reaches 0.4 g/L, the particle size of the product is uniform, which can well reduce the polarization of the electrode plate. The cell potential of electrolytic reaction is small, and the economic benefit is high. When the content of PEG is low, it has a low impact on the product and the economic benefits of electrolytic reaction. When the PEG content is higher than 0.4 g/L, the electrode reaction is hindered, resulting in an increase in cell potential.

## 1. Introduction

For a long time, the main application of the brine from potassium-magnesium-rich salt lakes has been to extract agricultural potassium fertilizer [1]. After potassium fertilizer is extracted, the high concentration brine crystallizes into bischofite by sunlight. The content of MgCl_2_·H_2_O in the bischofite is more than 96% [2]. It can be said that this kind of bischofite is an extremely high-quality magnesium resource. However, this magnesium resource has not been effectively utilized. It is mostly stacked around the salt lake, which will not only cause waste but also greatly damage the ecological resources of the salt lake [3].

With the shortage of resources and the improvement of human awareness of ecological resources protection, relevant experts and scholars began to focus on the treatment and application of bischofite resources. These research directions mainly include the preparation of metal magnesium by molten salt electrolysis and the preparation of Mg(OH)_2_ by alkaline neutralization [4,5,6]. Molten salt electrolysis requires dehydration of bischofite, which is difficult to industrialize, can be described as a worldwide problem [7,8]. In addition, the ability of this method to treat bischofite is very limited, so it is very unrealistic to solve the problem of excessive accumulation of bischofite. Compared with molten salt electrolysis, the cost and energy consumption of the alkaline neutralization method for preparing Mg(OH)_2_ are low, but this method is often very difficult to precipitate and filter, and the heterogeneity of purity and morphology often affects the application field of products.

Compared with the alkaline neutralization method, magnesium hydroxide prepared by the electrolytic method has higher purity, controllable morphology and faster sedimentation rate, which cannot be achieved by other methods [9]. Based on the above reasons, many experts and scholars began to devote themselves to researching the preparation of Mg(OH)_2_ by electrodeposition [10,11]. Kamath prepared Mg(OH)_2_ thin films on metal substrates by electrodeposition with Mg(NO_3_)_2_ solution and MgCl_2_ solution as raw materials, respectively, and investigated the effects of different process parameters on the electrodeposition product Mg(OH)_2_ [12,13,14]. Li et al. prepared Mg(OH)_2_ thin films by electrodeposition with Pt as substrate and Mg(NO_3_)_2_ as raw material and investigated the effect of deposition potential on the morphology of the films [15].

The reaction principle of electrolytic preparation of Mg(OH)_2_ is very similar to that of the alkaline-chlorination process preparation of NaOH [16,17]. Near the cathode plate, the OH^−^ produced by electrolysis combined with free Mg^2+^ in the solution and formed Mg(OH)_2_. Moreover, the cathode plate served as the substrate for the growth of Mg(OH)_2_ nuclei. When Mg(OH)_2_ crystals grow to a certain extent, the H_2_ produced by electrolytic reaction peels off these Mg(OH)_2_ from the surface of the electrode plate, resulting in the rapid settlement of the product [18,19]. After a certain time of electrolysis, the heterogeneous nucleation of Mg(OH)_2_ is very intense. A large number of Mg(OH)_2_ crystal nuclei will adhere to the surface of the plate, reducing the number of active sites on the substrate [20]. It will not only affect the rapid settlement of Mg(OH)_2_ but also increase the cell potential and reduce the current efficiency.

Polyethylene glycol (PEG) is a synthetic polymer with stable properties and good water solubility. It is widely used in medicine, paint, electroplating, pesticide, metal processing and food processing industries. Differences in PEG molecular weight often lead to differences in its physical and chemical properties. Among them, PEG-4000 is often used as a dispersant and emulsifier in industrial production to reduce product agglomeration. At the same time, PEG has strong water solubility. In the liquid medium, the ether bond in its molecule has a weak negative charge, which can be complexed with cations such as Ca^2+^ and Mg^2+^ in the solution. Therefore, the PEG macromolecule acts as a site for the growth of magnesium hydroxide crystals, which promotes the formation and sedimentation of Mg(OH)_2_.

In this study, Mg(OH)_2_ was prepared by electrolysis with bischofite aqueous solution as a magnesium source. PEG-4000 was innovatively introduced as a surfactant to enhance the sedimentation performance of Mg(OH)_2_ and accelerate the stripping of products from the substrate surface. The action mechanism of PEG-4000 in the preparation of Mg(OH)_2_ by electrolysis was analyzed for the first time.

## 2. Materials and Methods

### 2.1. Materials

In this experiment, bischofite was used as the raw material of electrolytic Mg(OH)_2_. The bischofite was dissolved and filtered to obtain a clear aqueous solution. The composition was analyzed by atomic emission spectrometer and an atomic absorption spectrometer. The composition of bischofite is shown in Table 1.

PEG-4000 was the additive and was purchased from Shanghai McLean Biochemical Technology Co., Ltd. All the chemical reagents were of analytical grade purity without further purification, and the water used in the progress of the experiment was deionized water.

### 2.2. Preparation of Mg(OH)_2_

The preparation of Mg(OH)_2_ by the electrolytic method was carried out in an electrolytic cell made of polyvinyl chloride. The device diagram is shown in Figure 1. The cathode plate and anode plate were made of platinum, with a dimension of 40 mm × 40 mm. The distance between them was controlled to be 50 mm. The device diagram is shown in Figure 1.

The basic principle of preparing Mg(OH)_2_ by electrolytic bischofite aqueous solution can be expressed by the following electrode reaction.
Anode: 2Cl^−^ → Cl_2_ + 2e^−^(1)
Cathode: 2H_2_O + 2e^−^ → 2OH^−^ + H_2_(2)

The cathode electrolyte was the filtered and diluted bischofite aqueous solution. However, there were some differences in the composition of anode electrolyte. During long-time electrolytic treatment, Cl_2_ produced in the anode chamber reacts with water to produce HCl and HClO. These by-products will penetrate into the cathode chamber, consume H^+^ and hinder the progress of the cathode reaction. Therefore, compared with the anode electrolyte, a certain amount of Mg(OH)_2_ slurry and H_2_O_2_ need to be introduced to neutralize the generated acid. The purpose of water resource recycling was realized by regularly replacing electrolyte.

The dissolved and filtered bischofite was used as the magnesium source, and the concentration of Mg^2+^ in the electrolyte was controlled to 0.35 mol/L. During the experiment, the current density was adjusted to 0.05 A/cm^2^, and the reaction temperature was 25 °C.

### 2.3. Characterization of Mg(OH)_2_

A scanning electron microscope was used to characterize the morphology of the product (SEM; Hitachi S-4800; Hitachi Seiki Ltd, Tokyo, Japan), of which the working potential of the scanning electron microscope was 2 kV. The particle size of the product was measured by a laser particle size analyzer (BT-9300HT; Dandong, China). Fourier transform infrared spectrometry spectrum was recorded by a Fourier transform infrared spectroscopy analysis (FT-IR; Nicolet 6700; Thermo Fisher Scientific; MA, U.S.). The phase composition of the product was characterized by an X-ray spectrometer (XRD; X-Pertpro; PANalytic B.V., Almelo, Netherlands), with Cu Kα1 radiation source, the excitation potential was 2.2 kV, the scanning range was 10°≤ 2θ ≤ 85°and the scanning speed was 0.2°/min.

The peeling rate represents the mass of magnesium hydroxide peeled off per square centimeter of the cathode substrate surface per unit time (1 h). It should be noted that the measurement of the stripping rate needs to be carried out after the mass transfer process has reached equilibrium. The peeling rate represents the mass of magnesium hydroxide peeled off per square centimeter of the cathode substrate surface per unit time (1 h). It should be noted that the measurement of the peeling rate needs to be carried out after the mass transfer process has reached equilibrium. Moreover, in order to ensure the accuracy of the measurement and not destroy the mass transfer balance, it is necessary to use a syringe to suck out the magnesium hydroxide generated before the mass transfer balance. After completion of the reaction, the obtained magnesium hydroxide was sufficiently washed and then dried. The peeling rate can be calculated by the following formula.
V = m/S(3)
where V is the peeling rate (g/cm^2^), m is the quality of Mg(OH)_2_, S is the surface area of electrode plate.

## 3. Result and Discussion

The trend of cell potential for electrolytic preparation of magnesium hydroxide is shown in Figure 2. The trend of cell potential under different conditions is basically the same. The cell potential value is larger in the initial time (600–100 s), which is attributed to the effect of the mass transfer process. When the reaction is carried out for a period of time (about 1000s), the cell potential value gradually stabilizes. This indicates that the mass transfer process in the electrolyzer has reached equilibrium at this time. After reaching the mass transfer equilibrium, the cell potential value showed an increasing trend under some conditions, relating to the decrease in the number of active sites on the substrate surface. The introduction of PEG-4000 has a significant effect on reducing and stabilizing the cell potential value. When the PEG-4000 concentration in the electrolyte was 0.2 g/L and 0.4 g/L, the cell potential value did not increase significantly after the mass transfer process reached equilibrium. In particular, when the PEG-4000 concentration was 0.4 g/L, the trend of cell potential decreased slightly and was significantly lower than that under other conditions. The energy consumption of electrolysis is relatively low, which means that the electrolytic reaction has better economic performance under this condition.

Figure 3 shows the relationship between the concentration of PEG-4000 and the peeling rate of magnesium hydroxide. It has been clearly seen that the peeling rate of Mg(OH)_2_ increases with the increase of PEG concentration initially. Magnesium hydroxide can be produced up to 0.055 g/cm^2^ per hour when the concentration of PEG-4000 is 0.4 g/L, which is the maximum under this system. When the PEG-4000 concentration in electrolyte is further increased, the peeling rate of Mg(OH)_2_ is significantly decreased. When the addition amount of PEG is 0.8 g/L and 1.0 g/L, the peeling speed of the product is lower than that of other conditions.

Figure 4 shows the particle size distribution of magnesium hydroxide prepared with different PEG-4000 addition. The relevant data can be obtained in Table 2. It can be found that the addition of PEG-4000 has a great influence on the particle size distribution of magnesium hydroxide. When the concentration of PEG-4000 is 0 g/L and 0.2 g/L, the particle size distribution of magnesium hydroxide is wide. When the concentration of PEG-4000 is 0.4 g/L, the particle size distribution of magnesium hydroxide is adjusted, and the distribution range is relatively concentrated, indicating that the particle size of magnesium hydroxide is relatively uniform at this time. The concentration of PEG-4000 in the electrolyte continued to increase, the particle size of magnesium hydroxide was larger and the degree of uniformity decreased significantly.

Figure 5 shows the X-ray diffraction pattern of the surface of the plate after electrodeposition. Compared with the standard pattern (ICSD #165674), it can be found that the substance deposited on the surface of the electrode is pure phase magnesium hydroxide. The XRD diffraction pattern of the obtained Mg(OH)_2_ crystal has sharp peaks, indicating that the Mg(OH)_2_ crystal has good crystallization properties. According to the spectrum, the product Mg(OH)_2_ has a stronger tendency to grow along the (101) crystal plane. This is understandable. In Mg(OH)_2_ crystal, the (101) plane is connected by a weak hydrogen bond and the (001) plane is connected by an ionic bond, so the (101) plane growth needs to break a higher potential barrier [21].

Figure 6 shows the SEM image of the electrode surface after the electrode reaction for 1 min in the mass transfer equilibrium state. The morphology was a highly dense and uniform continuous network of thin films with a thickness of 0.1–0.2 μm with no surfactant added (Figure 6a). According to the theory of non-uniform nucleation, the surface of the working electrode plays a catalytic role in nucleation so that the nucleus will preferentially form on the substrate. Under the impact of hydrogen bubbles on the surface of the substrate, these nuclei grow in the direction perpendicular to the plate and finally form porous Mg(OH)_2_ [22,23]. The strong alkaline environment at the cathode/electrolyte interface weakens the intralayer hydrogen bond at the edge of the Mg(OH)_2_ film, which leads to the product presenting a curly sheet shape [24]. In order to alleviate the stress caused by the asymmetric hydrogen bonds between layers, Mg(OH)_2_ particles curl and form the porous structure finally. Inevitably, this layer of magnesium hydroxide, which grows perpendicular to the surface of the plate, will be served as the substrate for secondary nucleation. It can be found that the Mg(OH)_2_ produced by secondary nucleation presents a flocculent shape under the condition of no PEG-4000 addition. Its existence will close the pores of network Mg(OH)_2_ and reduce the number of active sites on the substrate surface. It is the main reason for the rise of cell potential and the uneven particle size of Mg(OH)_2_. When the concentration of PEG-4000 was 0.2 g/L and 0.4 g/L, the Mg(OH)_2_ on the electrode surface still showed a dense and uniform continuous network morphology. The flocculent Mg(OH)_2_ produced by secondary nucleation disappears and is replaced by uniform flake products. When the concentration of PEG-4000 increased to 0.8 g/L, there was no significant change in the Mg(OH)_2_ initially grown on the surface of the electrode. Secondary crystallization occurs on the surface of the initial Mg(OH)_2_ crystal, and an irregular network structure with radial growth is obtained. When the PEG-4000 content in the electrolyte reaches 1.0 g/L, the surface morphology of the electrode is shown in Figure 6d. The morphology of Mg(OH)_2_ obtained from the nucleation of the electrode surface changed greatly. The electrode plate is covered by a grid-like distribution of Mg(OH)_2_ crystals and a film attached to the electrode surface in the grid, which is the factor that causes the non-uniform particle size distribution of the product. The film, which develops parallel to the plate surface makes the active sites on the plate surface reach the minimum, and the cell potential is also higher.

Figure 7 shows the SEM images of as-prepared Mg(OH)_2_. There is no significant change in the size of the Mg(OH)_2_ crystal in the range of PEG 4000 content of 0–0.4 g/L, which is consistent with the results measured by the laser particle size analyzer. Since the crystal tends to grow along the (101) crystal plane, the product exhibits a flake shape. When the amount of PEG-4000 added was 0.6 g/L and above, the surface area of magnesium hydroxide was significantly increased. This can be attributed to the fact that the PEG-4000 provides the growth context for magnesium hydroxide, which has a positive effect on the expansion of the (101) crystal plane. At the same time, as the particle size of the flaky product increases, the stress effect inside the product is more significant. The flaky magnesium hydroxide crystals are thus distorted.

The X-ray diffraction patterns of the products are shown in Figure 8. According to the XRD diffraction pattern shown in Figure 8, we obtained the peak intensity ratio of (101) crystal plane to (001) crystal plane (I_(101)/(001)_ values) of the samples prepared under different conditions, as shown in Table 3. It is worth mentioning that the value of I_(101)/(001)_ obtained from the standard atlas (ICSD #165674) is equal to 1.887. The I_(101)/(001)_ values of the obtained samples are significantly higher than that of the standard spectrum, indicating that the product is a flake product with the (101) crystal surface as the main exposed surface. It could be found that the I_(101)/(001)_ of sample c (The concentration of PEG equals 0.4 g/L) reached the minimum value of 2.061 among all samples. This means that the Mg(OH)_2_ has a high degree of crystallinity and low internal stress, as well as better dispersion properties [21]. This is mainly due to the large surface area of magnesium hydroxide, resulting in large internal stress in it.

Figure 9 shows the FT-IR diagram of Mg(OH)_2_ prepared under the condition of the PEG-4000 concentration of 0.8 g/L. The peak in the vicinity of 1088 cm^−1^ is the asymmetric expansion vibration peak of C-O-C in the PEG macromolecule, which indicates that the PEG-4000 is adsorbed on the Mg(OH)_2_ surface to a certain extent. The peaks at 2928 cm^−1^ and 2857 cm^−1^ belong to the asymmetric and symmetric vibrational peaks of -CH_2_, respectively. The peaks at 3440 cm^−1^ are derived from the hydroxyl vibrations of PEG [25]. The peaks at1458 cm^−1^, 1358 cm^−1^ and 3700 cm^−1^ are respectively -OH bond of a water molecule and the stretching vibration of Mg(OH)_2_. The peak at 437 cm^−1^ is the stretching vibration peak of the Mg-O skeleton.

According to the above results, it can be seen that the PEG molecular chain can serve as the base for the growth of Mg(OH)_2_ crystals (The ether group of the surfactant and the hydroxyl group of the Mg(OH)_2_ are bonded by hydrogen bonding.). When PEG-4000 was gradually added to the electrolyte until 0.4 g/L, the crystallization properties of magnesium hydroxide increased steadily. The product tends to grow along the (101) plane. When the PEG-4000 concentration reached 0.4 g/L, the particle size of the product did not increase significantly but was more uniform. The fine grains were significantly reduced, which was the main reason for the reduction of the cell potential and the rapid peeling of the product. When the concentration of PEG-4000 in the electrolyte reaches 0.6 g/L or above, PEG-4000 will be adsorbed on the surface of the electrode plate and hinder the occurrence of the electrode reaction. A large amount of Mg^2+^ forms complexes with PEG-4000 [26]. However, because the electrode reaction is hindered, the generation of OH^−^ is also limited. According to Pauling’s first rule, the Mg(OH)_6_^4−^ negative ion formed by Mg^2+^ and OH^−^ is an effective structural unit for the formation of flaky magnesium hydroxide [27]. The lower rate of OH^−^ generation leads to the existence of magnesium hydroxide, mostly with small nuclei or even amorphous, which further hinders the occurrence of electrode reaction. Since the electrode reaction is inhibited, there will be no hydrogen at these sites to exfoliate the electrode surface products in time. These reasons also lead to the high cell potential of the electrolysis reaction and the difficulty of peeling off the products on the substrate surface.

## 4. Conclusions

(1) The ether group of the PEG molecular chain and the hydroxyl group of Mg(OH)_2_ can be combined by hydrogen bonding, which provides a "template" for the growth of magnesium hydroxide. Therefore, PEG plays an important role in regulating the growth mode, morphology and sedimentation performance of Mg(OH)_2_ crystals.

(2) When the PEG concentration is low (no higher than 0.4 g/L), the crystallization and sedimentation properties of magnesium hydroxide are gradually optimized with the introduction of PEG. When the PEG content reaches 0.4 g/L, the grain size of the product is uniform, which can well reduce the polarization of the polar plate. The cell potential of the electrolysis reaction is smaller, and the economic benefit is higher.

(3) Under the higher introduction amount, the role of PEG is more prominent in the aspect of crystal growth context and the promotion of crystal particle size increase. At the same time, it will also hinder the occurrence of electrode reaction, resulting in the rise of cell potential.

## Figures and Tables

**Figure 1 materials-15-03278-f001:**
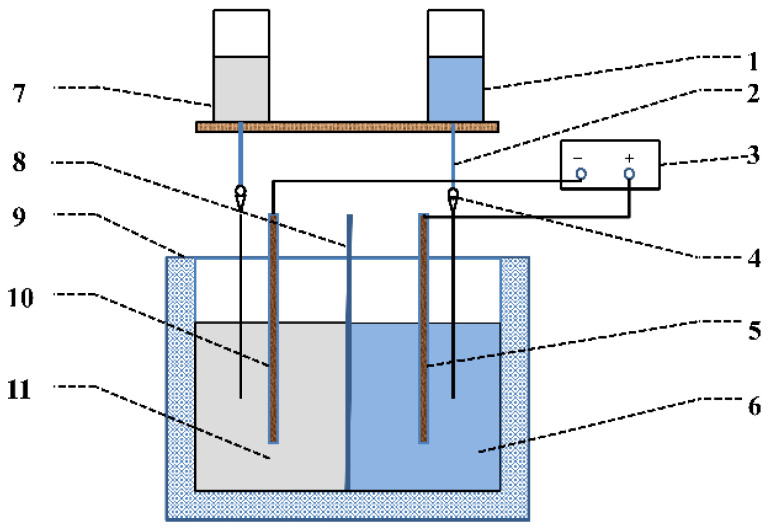
Experimental device diagram. (1)—Cathode electrolyte, (2)—Feeding pipe, (3)—DC power, (4)—Rotameter, (5)—Cathode plate, (6)—Cathode chamber, (7)—Anode electrolyte, (8)—Diaphragm, (9)—Electrolyzer, (10)—Anode plate, (11)—Anode chamber.

**Figure 2 materials-15-03278-f002:**
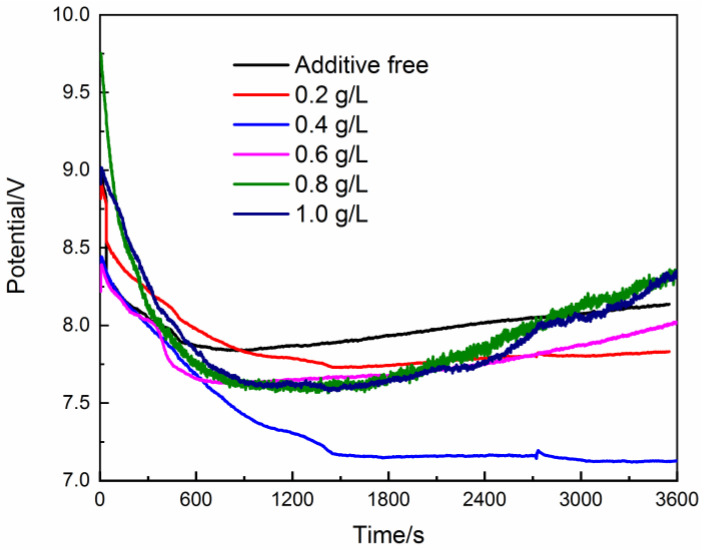
Cell potential curve of electrolytic reaction.

**Figure 3 materials-15-03278-f003:**
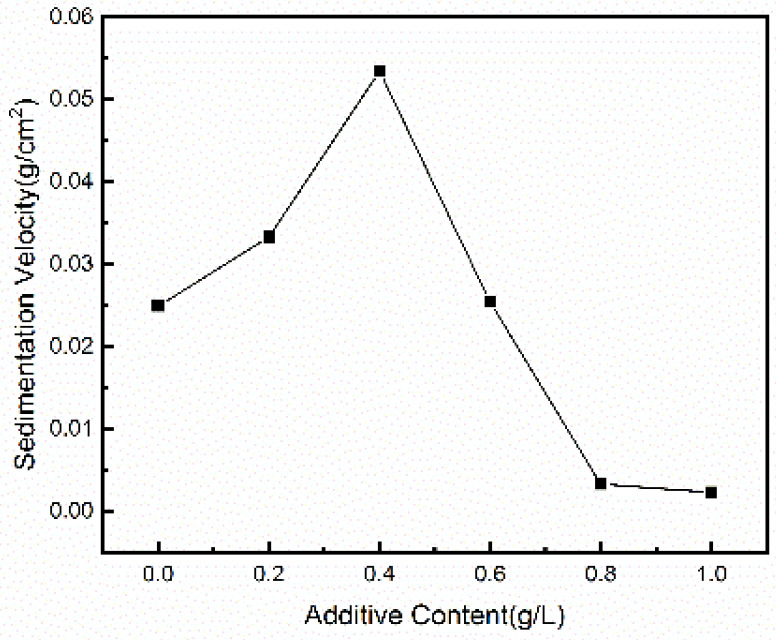
Peeling rate of as-prepared Mg(OH)_2_.

**Figure 4 materials-15-03278-f004:**
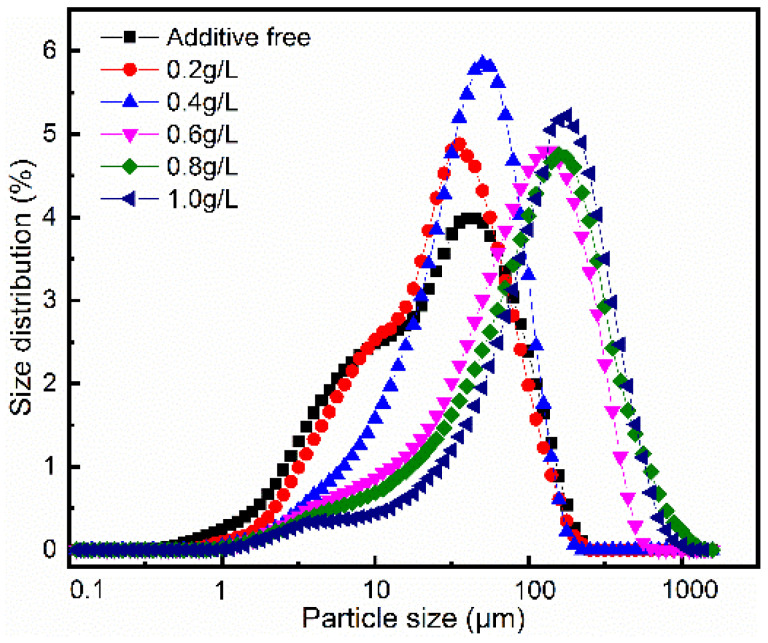
Grain size distribution curve of Mg(OH)_2_.

**Figure 5 materials-15-03278-f005:**
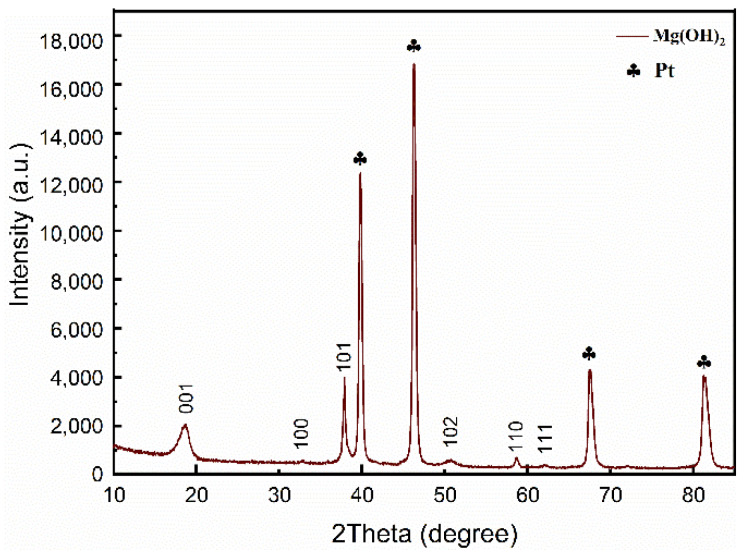
Powder X-ray diffraction patterns of as-prepared Mg(OH)_2_ on electrode surface.

**Figure 6 materials-15-03278-f006:**
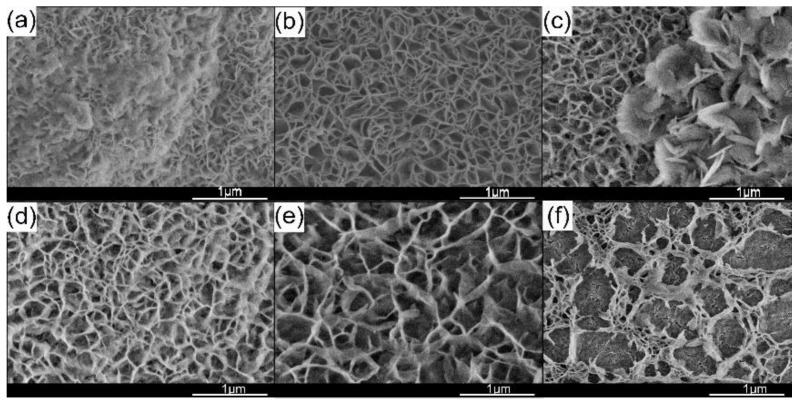
SEM images of as-prepared Mg(OH)_2_ on electrode surface. (**a**) Additive free; (**b**) 0.2 g/L; (**c**) 0.4 g/L; (**d**) 0.6 g/L; (**e**) 0.8 g/L; (**f**) 1.0 g/L.

**Figure 7 materials-15-03278-f007:**
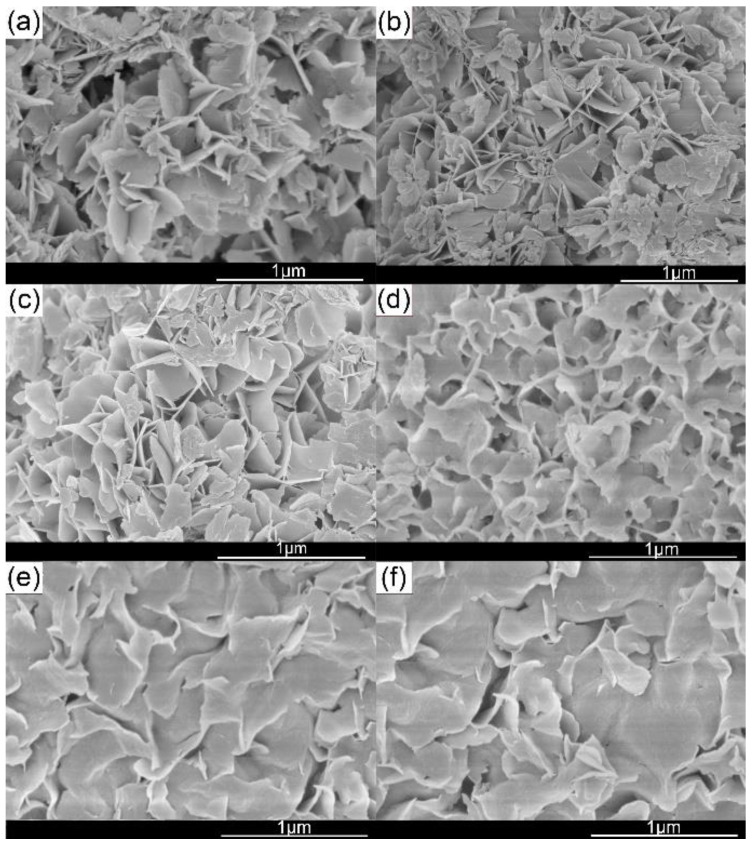
SEM images of as-prepared Mg(OH)_2_. (**a**) Additive free; (**b**) 0.2 g/L; (**c**) 0.4 g/L; (**d**) 0.6 g/L; (**e**) 0.8 g/L; (**f**) 1.0 g/L.

**Figure 8 materials-15-03278-f008:**
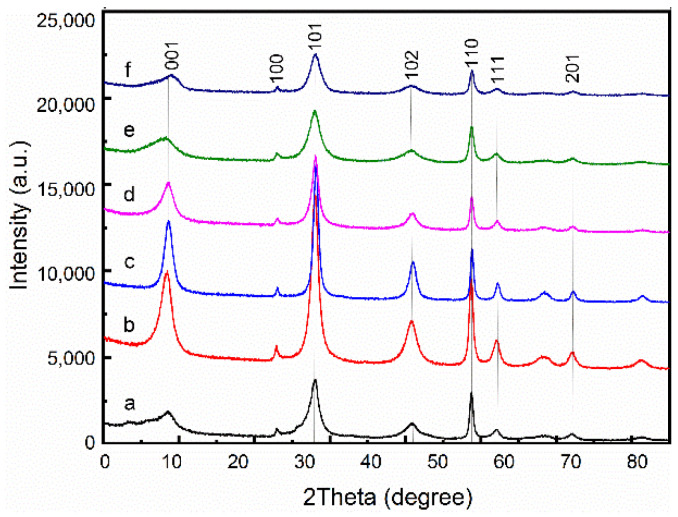
X-ray diffraction patterns images of as-prepared Mg(OH)_2_. (**a**) Additive free; (**b**) 0.2 g/L; (**c**) 0.4 g/L; (**d**) 0.6 g/L; (**e**) 0.8 g/L; (**f**) 1.0 g/L.

**Figure 9 materials-15-03278-f009:**
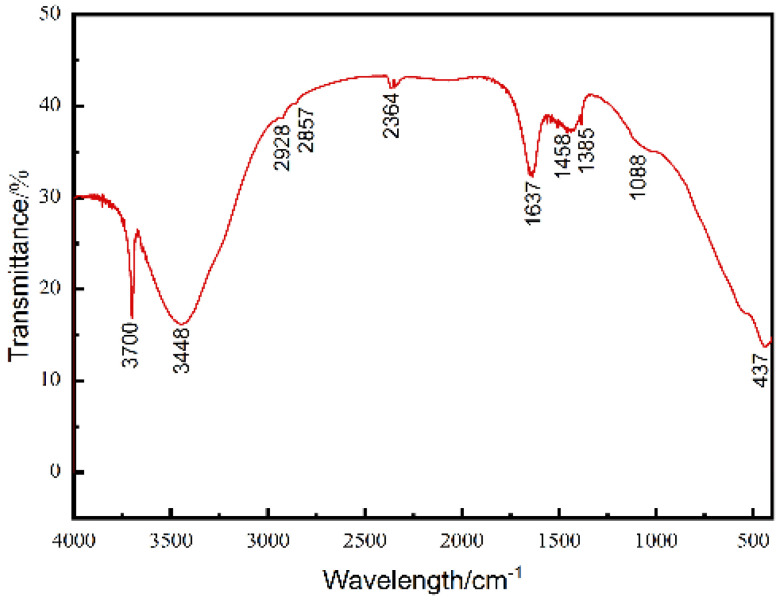
FT-IR spectra of as-produced Mg(OH)_2_.

**Table 1 materials-15-03278-t001:** The composition analysis of bischofite.

Composition	Concentration (g/L)	Composition	Concentration (g/L)
Mg^2+^	54.40	BO_3_^2−^	2.81 × 10^−3^
Ca^2+^	0.46	Cu^2+^	8.15 × 10^−5^
Na^+^	1.21	K^+^	0.63
SO_4_^2−^	1.20	Li^+^	0.14
Cl^−^	177.48	Mn^2+^	5.24 × 10^−5^
Fe^3+^	1.42 × 10^−4^	Ni^2+^	1.41 × 10^−5^
Al^3+^	2.13 × 10^−4^	PO_4_^3−^	4.76 × 10^−5^
Si^4+^	3.45 × 10^−4^	Pb^2+^	7.49 × 10^−5^

**Table 2 materials-15-03278-t002:** Particle size distribution data of electrolytic products with different PEG addition.

PEG Content (g/L)	Additive Free	0.2	0.4	0.6	0.8	1.0
D_50_ ^1^	25.5	21.7	20.0	28.8	37.5	52.2
D_90_ ^1^	58.0	59.0	60.7	68.6	93.0	121.0

^1^ D50 and D90 were the corresponding particle sizes when the cumulative particle size distribution percentage of the sample reaches 50% and 90%, respectively.

**Table 3 materials-15-03278-t003:** The peak intensity ratio of (101) crystal plane to (001) crystal plane of electrolytic products with different PEG-4000 addition.

PEG Content (g/L)	Additive Free	0.2	0.4	0.6	0.8	1.0
I_(101)/(001)_	1.972	2.041	2.061	2.058	2.030	2.033

## Data Availability

Not applicable.

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
