# Peer review of "Effect of Polyethylene Glycol on Preparation of Magnesium Hydroxide by Electrodeposition"

_materials, 2022, doi:10.3390/ma15093278_

Round 1

Reviewer 1 Report

This article reports the influence of polyethylene glycol on the preparation of magnesium hydroxide by electrolysis from bischofite. This research might be of interest to the community interested in Mg(OH)2 particles with controllable morphology. However, there are many inconsistencies in the text and data interpretations which must be improved before publishing this manuscript in the Journal: Materials

The comments for the authors are provided below.

From “Preparation of Mg(OH)2” part (line 79) and from Fig. 1 it is unclear what was used for the anode and cathode material. Also, what was anode electrolyte? This should be carefully revised and defined in text as well as in Fig. 1.

Line 84: (5) – it is written “cathode electrolyte”, it should be corrected to “cathode plate” or whatever cathode material they used.

They stated that PEG molecular weight 4000 was used, but in the table (Line 135-136) we have two different PEG numbers used. The authors need to define what labels “D50” and “D90” mean.

Line 102 and 108:  Figure 1 should be corrected to Figure 2.

Line 103-104: The meaning of the sentence “In the initial period of time (600-100s), the cell potential is affected by the mass transfer process” is unclear, especially since Fig. 2 shows values only ​​up to 60 s. Likewise, the sentence (Line 109-112) “When the PEG concentration in the electrolyte is 0.2 g/L and 0.4 g/L, the cell potential does not increase significantly after reaching the minimum value. Especially when the PEG concentration is 0.4 g/L, the trend of cell potential decreases slightly and is significantly lower than that under other conditions.” should be carefully revised.

The Authors did not discuss effect of hydrogen evolution on the process of electrolysis and formed morphology (please see in http://dx.doi.org/10.1016/j.apsusc.2011.04.098, https://doi.org/10.2298/JSC180913084V, https://doi.org/10.1016/j.electacta.2018.02.121). Especially in the situation as represent in Fig. 6 where honey-comb like structures are formed. They should state the difference between Fig. 6 and Fig. 7. In Fig. 6 the bars are not visible and clear enough, please correct and remove unnecessary data from SEM images.

Line 181: please give the size of particle analyzed.

Lines 225-226: The references 22-24 should be cited in the Introduction part, because they deal with Mg(OH)2 deposition, without PEG.

Also, there are, in main text, many typos the Authors need to make appropriate corrections throughout the whole manuscript.

Author Response

List of Responses

Dear Editors and Reviewers:

Thank you for your letter and for the reviewers’ comments concerning our manuscript entitled “Effect of Polyethylene Glycol on Preparation of Magnesium Hydroxide by Electrodeposition”. (ID: materials-1695947). Those comments are all valuable and very helpful for revising and improving our paper, as well as the important guiding significance to our researches. We have studied comments carefully and have made correction which we hope meet with approval. Revised portion are marked by track changes function in the paper. The main corrections in the paper and the responds to the reviewer’s comments are as flowing:

Responds to the reviewer’s comments:

  1. Response to comment: From “Preparation of Mg(OH)2” part (line 79) and from Fig. 1 it is unclear what was used for the anode and cathode material. Also, what was anode electrolyte? This should be carefully revised and defined in text as well as in Fig. 1.

Response: The cathode plate and anode plate were made of platinum, with a dimension of 40 mm×40mm. The distance between them was controlled to be 50mm.

The basic principle of preparing Mg(OH)2 by electrolytic bischofite aqueous solution can be expressed by the following electrode reaction.

Anode: 2Cl-→Cl2+2e-

(1)

Cathode: 2H2O+2e-→2OH-+H2

(2)

The cathode electrolyte was the filtered and diluted bischofite aqueous solution. However, there were some differences in the composition of anode electrolyte. During long-time electrolytic treatment, Cl2 produced in the anode chamber reacts with water to produce HCl and HClO. These by-products will penetrate into the cathode chamber, con-sume H+, and hinder the progress of cathode reaction. Therefore, compared with the anode electrolyte, a certain amount of Mg(OH)2 slurry and H2O2 need to be introduced to neutralize the generated acid. The purpose of water resource recycling was realized by regularly replacing electrolyte.

  1. Response to comment: Line 84: (5) – it is written “cathode electrolyte”, it should be corrected to “cathode plate” or whatever cathode material they used.

Response: The cathode plate and anode plate were made of platinum, with a dimension of 40 mm ×40mm. The distance between them was controlled to be 50mm. The above statement has been given in the text.

  1. Response to comment: They stated that PEG molecular weight 4000 was used, but in the table (Line 135-136) we have two different PEG numbers used. The authors need to define what labels “D50” and “D90” mean.

Response: D50 and D90 were the corresponding particle sizes when the cumulative particle size distribution percentage of the sample reaches 50% and 90%, respectively. In order to make the content of this part more clear, we have added a note below the table.

  1. Response to comment: Line 102 and 108: Figure 1 should be corrected to Figure 2.

Response: In response to this suggestion, we have revised the original content.

  1. Response to comment: Line 103-104: The meaning of the sentence “In the initial period of time (600-100s), the cell potential is affected by the mass transfer process” is unclear, especially since Fig. 2 shows values only ​​up to 60 s. Likewise, the sentence (Line 109-112) “When the PEG concentration in the electrolyte is 0.2 g/L and 0.4 g/L, the cell potential does not increase significantly after reaching the minimum value. Especially when the PEG concentration is 0.4 g/L, the trend of cell potential decreases slightly and is significantly lower than that under other conditions.” should be carefully revised.

Response: Due to our mistake, the units of the abscissa axis of Figure 2 are filled in wrong. We have reworked Figure 2 and revised the relevant textual content.

  1. Response to comment: The Authors did not discuss effect of hydrogen evolution on the process of electrolysis and formed morphology (please see in http://dx.doi.org/10.1016/j.apsusc.2011.04.098, https://doi.org/10.2298/JSC180913084V, https://doi.org/10.1016/j.electacta.2018.02.121). Especially in the situation as represent in Fig. 6 where honey-comb like structures are formed. They should state the difference between Fig. 6 and Fig. 7. In Fig. 6 the bars are not visible and clear enough, please correct and remove unnecessary data from SEM images.

Response: We have discussed how hydrogen acts in electrodeposition as suggested and incorporated the above article as a reference. Figure 6 shows the SEM image of the electrode surface after the electrode reaction for 1 min in the mass transfer equilibrium state. It is used to observe the growth state of the product on the surface of the substrate. Figure 7 shows an SEM image of magnesium hydroxide peeled off from the substrate surface. We re-described these differences in the text.

  1. Response to comment: Line 181: please give the size of particle analyzed.

Response: The particle size of the product has been given in Figure 4 and Table 2.

  1. Response to comment: Lines 225-226: The references 22-24 should be cited in the Introduction part, because they deal with Mg(OH)2 deposition, without PEG.

Response: According to the structure and content of the article, we adjusted the citation position and order of references.

  1. Response to comment: Also, there are, in main text, many typos the Authors need to make appropriate corrections throughout the whole manuscript.

Response: According to this advice, we reviewed the article and corrected some grammatical and typing errors.

Reviewer 2 Report

The tackling of environmental issues towards a more sustainable waste management and manufacturability is of utmost importance, whereon the materials play a major role. This work intends to study the impact of several PEG additions to the electro-crystallisation of magnesium hydroxide as a recovery method. In my opinion, the topic is relevant for the journal and the manuscript is solid. Nevertheless, I have some questions and suggestions for improvement, as follows.

  1. In line 55, do the authors mean “the OH is produced”?
  2. In line 60, there is a period that should be a comma to connect the two sentences.
  3. The authors used PEG-4000. What was the criteria for the use of such molecular weight? Would authors expect different results using different molecular weight? Perhaps, authors could comment on the reasons why they chose PEG-4000.
  4. In Figure 1, there is relevant information missing. Authors should indicate the Materials and Methods the shape and surface area of the electrodes (both anode and cathode), and their composition (platinum?). Additionally, what is the anode electrolyte composition? Could authors present and describe each of the half-cell reactions?
  5. Was the electrodeposition conducted galvanostatically? If so, authors should indicate the current density applied.
  6. In line 99, the scanning range is presented in terms of theta, but the X-ray diffractograms of figures 5 and 8 indicate 2-theta.
  7. How was the peeling rate measured?
  8. The authors mention a maximum of 0.055 g per hour of magnesium hydroxide production. Is this amount reasonable for industrial application? Perhaps, authors could comment on that.
  9. In line 120, authors comment on Figure 3 and state that the peeling rate of magnesium hydroxide is not significantly increased above 0.4 g/L. However, Figure 3 shows clearly that the peeling rate decreases as the PEG concentration increases beyond 0.4 g/L.
  10. The authors put some efforts in characterising the particle growth and size, but it is not clear the relevance for that. How do particle size and crystallographic growth direction impact applicability and process relevance?
  11. In Table 2, there is a Chinese word that should be translated (perhaps ‘content’?).
  12. In line 140, it is stated the intensity ratio of peaks (101)/(001) is much higher than the standard. Authors should indicate the intensity ratio of the standard for those sets of planes, for proper comparison.
  13. The sentence on the preferred growth plane in line 141 is confusing.
  14. Line 146, unnecessary use of capital letter in “magnesium”.
  15. Line 147, unnecessary use of capital letter in “morphology”.
  16. Line 146, micrometres should have proper units: μm.
  17. In line 159 (and in other instances), “peg” should be written as “PEG”.
  18. In Figure 6, the scale bars are not clearly visible. Also, magnification should be the same for all the instances in Figure 6.
  19. In the Conclusions, authors mention that by using 0.4 g/L of PEG “the economic benefit is higher”, but it is not explained why. What aspects of the process make it economically more beneficial? This could probably be mentioned earlier in the manuscript.
  20. I recommend a careful proofreading of the manuscript to correct syntax and typos.

Author Response

List of Responses

Dear Editors and Reviewers:

Thank you for your letter and for the reviewers’ comments concerning our manuscript entitled “Effect of Polyethylene Glycol on Preparation of Magnesium Hydroxide by Electrodeposition”. (ID: materials-1695947). Those comments are all valuable and very helpful for revising and improving our paper, as well as the important guiding significance to our researches. We have studied comments carefully and have made correction which we hope meet with approval. Revised portion are marked by track changes function in the paper. The main corrections in the paper and the responds to the reviewer’s comments are as flowing:

Responds to the reviewer’s comments:

  1. Response to comment: In line 55, do the authors mean “the OH is produced”?

Response: The basic principle of preparing Mg(OH)2 by electrolytic magnesium chloride aqueous solution can be expressed by the following electrode reaction.

Anode: 2Cl-→Cl2+2e-

Cathode: 2H2O+2e-→2OH-+H2

From the above reaction, OH- is indeed produced.

  1. Response to comment: In line 60, there is a period that should be a comma to connect the two sentences.

Response: As the sentence is too long to understand well, we have rewritten it.

  1. Response to comment: The authors used PEG-4000. What was the criteria for the use of such molecular weight? Would authors expect different results using different molecular weight? Perhaps, authors could comment on the reasons why they chose PEG-4000.

Response: Polyethylene glycol (PEG) is a synthetic polymer with stable properties and good water solubility. It is widely used in medicine, paint, electroplating, pesticide, metal processing and food processing industries. Differences in PEG molecular weight often lead to differences in its physical and chemical properties. Among them, PEG-4000 is often used as a dispersant and emulsifier in industrial production to reduce product agglomeration. At the same time, PEG has strong water solubility. In the liquid medium, the ether bond in its molecule has a weak negative charge, which can be complexed with cations such as Ca2+ and Mg2+ in the solution. Therefore, the PEG macromolecule acts as a site for the growth of magnesium hydroxide crystals, which promotes the formation and sedimentation of magnesium hydroxide.

Based on the above reasons, we chose PEG as the additive for the electrolysis reaction.

  1. Response to comment: In Figure 1, there is relevant information missing. Authors should indicate the Materials and Methods the shape and surface area of the electrodes (both anode and cathode), and their composition (platinum?). Additionally, what is the anode electrolyte composition? Could authors present and describe each of the half-cell reactions?

Response: The cathode plate and anode plate were made of platinum, with a dimension of 40 mm ×40mm. The distance between them was controlled to be 50mm.

The basic principle of preparing Mg(OH)2 by electrolytic bischofite aqueous solution can be expressed by the following electrode reaction.

Anode: 2Cl-→Cl2+2e-

(1)

Cathode: 2H2O+2e-→2OH-+H2

(2)

The cathode electrolyte was the filtered and diluted bischofite aqueous solution. However, there were some differences in the composition of anode electrolyte. During long-time electrolytic treatment, Cl2 produced in the anode chamber reacts with water to produce HCl and HClO. These by-products will penetrate into the cathode chamber, con-sume H+, and hinder the progress of cathode reaction. Therefore, compared with the anode electrolyte, a certain amount of Mg(OH)2 slurry and H2O2 need to be introduced to neutralize the generated acid. The purpose of water resource recycling was realized by regularly replacing electrolyte.

  1. Response to comment: Was the electrodeposition conducted galvanostatically? If so, authors should indicate the current density applied

Response: The electrodeposition was conducted galvanostatically, the current density was 0.05A/cm2. The above information has been given in the Preparation of Mg(OH)2 section

  1. Response to comment: In line 99, the scanning range is presented in terms of theta, but the X-ray diffractograms of figures 5 and 8 indicate 2-theta.

Response: Here we want to express the range of 2θ, so we changed θ to 2θ.

  1. Response to comment: How was the peeling rate measured?

Response: The peeling rate represents the mass of magnesium hydroxide peeled off per square centimeter of the cathode substrate surface per unit time (1 hour). It should be noted that the measurement of the stripping rate needs to be carried out after the mass transfer pro-cess has reached equilibrium. The peeling rate represents the mass of magnesium hy-droxide peeled off per square centimeter of the cathode substrate surface per unit time (1 hour). It should be noted that the measurement of the peeling rate needs to be carried out after the mass transfer process has reached equilibrium. And in order to ensure the accu-racy of the measurement and not destroy the mass transfer balance, it is necessary to use a syringe to suck out the magnesium hydroxide generated before the mass transfer balance. After completion of the reaction, the obtained magnesium hydroxide was sufficiently washed and then dried. The peeling rate can be calculated by the following formula.

V=m/S   (3)

Where V is the peeling rate (g/cm2), m is the quality of Mg(OH)2, S is the surface area of electrode plate.

  1. Response to comment: The authors mention a maximum of 0.055 g per hour of magnesium hydroxide production. Is this amount reasonable for industrial application? Perhaps, authors could comment on that.

Response: The electrolytic reaction adopts constant current treatment, and the current density is 0.05A per square centimeter.

The theoretical output (m0) of magnesium hydroxide per square centimeter of plate surface can be calculated by Faraday's second law. The transfer of 1mol of electrons in one hour is 26.8 A×h, and the transfer of 2 mol of electrons is required to generate 1 mol of magnesium hydroxide. The following formula can be obtained.

m0=M´I´t/(2´26.8)

In the above formula, M is the relative molecular weight of magnesium hydroxide, 58. I is the current intensity; t is the electrolysis time.

After calculation, the theoretical output of magnesium hydroxide should be about 0.054. Since the test of stripping rate needs to be carried out after the mass transfer reaction reaches equilibrium, and the products on the electrode plate surface are not removed in order to avoid damaging the equilibrium, it is possible that the actual measured value is higher than the theoretical value.

  1. Response to comment: In line 120, authors comment on Figure 3 and state that the peeling rate of magnesium hydroxide is not significantly increased above 0.4 g/L. However, Figure 3 shows clearly that the peeling rate decreases as the PEG concentration increases beyond 0.4 g/L.

Response: In response to this problem, we re-describe and analyze the phenomenon reflected in Figure 3.

  1. Response to comment: The authors put some efforts in characterising the particle growth and size, but it is not clear the relevance for that. How do particle size and crystallographic growth direction impact applicability and process relevance?

Response: In the process of preparing magnesium hydroxide by electrodeposition, the magnesium hydroxide in microcrystalline and amorphous state adheres to the surface of the substrate, which leads to an increase in cell voltage and an increase in production cost. This paper hopes to achieve better economics of the electrolysis process by introducing PEG to normalize the growth of the product, magnesium hydroxide. When the amount of PEG introduced was low, the energy consumption of the electrolysis reaction and the particle size of the product did not change significantly. When the introduction amount of PEG reaches 0.4 g/L, the particle size of the product is uniform and can be quickly peeled off the surface of the substrate, which makes the economic performance of the electrolysis reaction more prominent. With the further introduction of PEG, the complexation between PEG and Mg2+ is obvious, which not only leads to the deterioration of particle size uniformity, but also leads to the increase of cell voltage. As for the influence of grain growth direction and grain size on applicability, we will really conduct further research on this content in the future.

  1. Response to comment: In Table 2, there is a Chinese word that should be translated (perhaps ‘content’?).

Response: Thank you for your reminder. We have corrected this error.

  1. Response to comment: In line 140, it is stated the intensity ratio of peaks (101)/(001) is much higher than the standard. Authors should indicate the intensity ratio of the standard for those sets of planes, for proper comparison.

Response: We have revised the text of this part. The peak intensity ratio of each product under corresponding conditions is listed in the paragraph at the top of Figure 8 and Table 3.

  1. Response to comment: The sentence on the preferred growth plane in line 141 is confusing.

Response: We have revised the text of this part.

  1. Response to comment: Line 146, unnecessary use of capital letter in “magnesium”;

Line 147, unnecessary use of capital letter in “morphology”; Line 146, micrometres should have proper units: μm; In line 159 (and in other instances), “peg” should be written as “PEG”.

Response: We have checked and changed the full text according to the above suggestions.

  1. Response to comment: In Figure 6, the scale bars are not clearly visible. Also, magnification should be the same for all the instances in Figure 6.

Response: In order to ensure that the SEM pictures have the same magnification, we re-selected the pictures and marked the scale.

  1. Response to comment: In the Conclusions, authors mention that by using 0.4 g/L of PEG “the economic benefit is higher”, but it is not explained why. What aspects of the process make it economically more beneficial? This could probably be mentioned earlier in the manuscript.

Response: Compared with conventional methods, the main factor affecting the economy of magnesium hydroxide prepared by electrolytic method is power consumption. The consumption of electric energy can be reflected by the change of cell voltage. The higher the cell voltage, the higher the power consumption. This is understandable and can be found in our analysis of Figure 2.

  1. Response to comment: I recommend a careful proofreading of the manuscript to correct syntax and typos.

Response: According to this advice, we reviewed the article and corrected some grammatical and typing errors.

Reviewer 3 Report

The title is interesting regarding using polyethylene glycol (PEG) in the preparation of magnesium hydroxide by electrolysis. The findings are interesting and relevant to the growing interest in electrodeposition process. It can be published in journal of materials after minor revision.

  • Improve the abstract based on the results.
  • The novelty of the work should be emphasis. Pleas clarify the new idea of the work at the end of introduction.
  • Figures: brief description on figures required.

Figure 1: the quality is very low and the mentioned parts are not clear. It should be improved.

Author Response

List of Responses

Dear Editors and Reviewers:

Thank you for your letter and for the reviewers’ comments concerning our manuscript entitled “Effect of Polyethylene Glycol on Preparation of Magnesium Hydroxide by Electrodeposition”. (ID: materials-1695947). Those comments are all valuable and very helpful for revising and improving our paper, as well as the important guiding significance to our researches. We have studied comments carefully and have made correction which we hope meet with approval. Revised portion are marked by track changes function in the paper. The main corrections in the paper and the responds to the reviewer’s comments are as flowing:

Responds to the reviewer’s comments:

  1. Response to comment: Improve the abstract based on the results.

Response: According to your suggestion, we revised the abstract of the article according to the content in the conclusion.

  1. Response to comment: Improve the abstract based on the results.

Response: According to your suggestion, we revised the abstract of the article according to the content in the conclusion.

  1. Response to comment: The novelty of the work should be emphasis. Pleas clarify the new idea of the work at the end of introduction.

Response: The novelty of this paper mainly includes three aspects. Firstly, we used the aqueous solution of bischofite as magnesium source for the first time, and prepared magnesium hydroxide by electrolysis. Secondly, the growth mode of magnesium hydroxide prepared by electrodeposition was analyzed. Finally, the mechanism of PEG in the preparation of magnesium hydroxide by electrodeposition is introduced in detail. The above contents are introduced at the end of the introduction.

  1. Figures: brief description on figures required. Figure 1: the quality is very low and the mentioned parts are not clear. It should be improved.

Response: We introduced the description of the picture in Preparation of Mg(OH)2 section. And we reintroduce the high-resolution image of Figure 1.

Round 2

Reviewer 1 Report

Just minor correction, they cited one and the same work under two different records as [19] and [23], that should be corrected. As for the rest, the authors took into account all the comments and suggestions and made the necessary changes, so that I recommend the revised manuscript entitled “Effect of Polyethylene Glycol on Preparation of Magnesium Hydroxide by Electrodeposition” for publication.

Author Response

List of Responses

Dear Editors:

Thank you for your letter and for the reviewers’ comments concerning our manuscript entitled “Effect of Polyethylene Glycol on Preparation of Magnesium Hydroxide by Electrodeposition”. (ID: materials-1695947). Those comments are all valuable and very helpful for revising and improving our paper, as well as the important guiding significance to our researches. We have studied comments carefully and have made correction which we hope meet with approval. Revised portion are marked by track changes function in the paper. The main corrections in the paper and the responds to the reviewer’s comments are as flowing:

Responds to the reviewer’s comments:

  1. Response to comment: Just minor correction, they cited one and the same work under two different records as [19] and [23], that should be corrected. As for the rest, the authors took into account all the comments and suggestions and made the necessary changes, so that I recommend the revised manuscript entitled “Effect of Polyethylene Glycol on Preparation of Magnesium Hydroxide by Electrodeposition” for publication.

Response: Thank you for your suggestion. We proofread the references again.

Reviewer 2 Report

Figure 7e has a different scale length relative to the other instances. It should be corrected to have the same magnification and same scale bar.

Author Response

List of Responses

Dear Editors and Reviewers:

Thank you for your letter and for the reviewers’ comments concerning our manuscript entitled “Effect of Polyethylene Glycol on Preparation of Magnesium Hydroxide by Electrodeposition”. (ID: materials-1695947). Those comments are all valuable and very helpful for revising and improving our paper, as well as the important guiding significance to our researches. We have studied comments carefully and have made correction which we hope meet with approval. Revised portion are marked by track changes function in the paper. The main corrections in the paper and the responds to the reviewer’s comments are as flowing:

Responds to the reviewer’s comments:

  1. Response to comment: Figure 7e has a different scale length relative to the other instances. It should be corrected to have the same magnification and same scale bar.

Response: We introduced SEM images with the same magnification and re labeled the scale.